# Structural order enhances charge carrier transport in self-assembled Au-nanoclusters

Florian Fetzer[1,5], Andre Maier[2,3,5], Martin Hodas[4], Olympia Geladari[2,3], Kai Braun[2,3], Alfred J. Meixner[2,3], Frank Schreiber [3,4], Andreas Schnepf [1✉] & Marcus Scheele [2,3✉]

The collective properties of self-assembled nanoparticles with long-range order bear immense potential for customized electronic materials by design. However, to mitigate the shortcoming of the finite-size distribution of nanoparticles and thus, the inherent energetic disorder within assemblies, atomically precise nanoclusters are the most promising building blocks. We report an easy and broadly applicable method for the controlled self-assembly of atomically precise $Au_{32}(^nBu_3P)_{12}Cl_8$ nanoclusters into micro-crystals. This enables the determination of emergent optoelectronic properties which resulted from long-range order in such assemblies. Compared to the same nanoclusters in glassy, polycrystalline ensembles, we find a 100-fold increase in the electric conductivity and charge carrier mobility as well as additional optical transitions. We show that these effects are due to a vanishing energetic disorder and a drastically reduced activation energy to charge transport in the highly ordered assemblies. This first correlation of structure and electronic properties by comparing glassy and crystalline self-assembled superstructures of atomically precise gold nanoclusters paves the way towards functional materials with novel collective optoelectronic properties.

[1] Institut für Anorganische Chemie Universität Tübingen, Auf der Morgenstelle 18, D-72076 Tübingen, Germany. [2] Institut für Physikalische und Theoretische Chemie, Universität Tübingen, Auf der Morgenstelle 18, D-72076 Tübingen, Germany. [3] Center for Light-Matter Interaction, Sensors & Analytics Lisa+, Universität Tübingen, Auf der Morgenstelle 15, D-72076 Tübingen, Germany. [4] Institut für Angewandte Physik, Universität Tübingen, Auf der Morgenstelle 10, D-72076 Tübingen, Germany. [5]These authors contributed equally: Florian Fetzer, Andre Maier. ✉email: andreas.schnepf@uni-tuebingen.de; marcus.scheele@uni-tuebingen.de

Using the collective properties of self-assembled molecules and particles as building blocks bears immense opportunities for microelectronic applications[1–3]. Already implemented applications of self-assembled thin films range from light-emitting diodes (LED) over field-effect transistors (FET) to optical sensors[4]. Inorganic nanoparticles, organic π-systems, and conjugated polymers are the most widely used components for such self-assembly[5–8]. For instance, previous studies have shown the possibility to form three-dimensional assemblies with long-range order using gold nanoparticles as building blocks[9]. However, these nanoparticles consist of a few hundred to thousands of atoms, are not atomically precise, exhibit finite-size distributions, and thus, an inherent energetic disorder in ensembles. To mitigate this shortcoming, atomically precise, inorganic molecular clusters have been suggested as promising building blocks for customized electronic materials by design of their structure[10–13]. These materials exhibit larger dielectric constants than organic semiconductors with profound consequences for their excited-state properties, such as the ability to exploit quantum confinement effects. A variety of such molecular clusters, often referred to as superatoms, has already been used for the formation of solid-state materials[14–18]. A special interest is thereby focused on the influence of the structure of the assembled materials onto their properties, possibly enabling the creation of materials with desired properties by design[19,20].

Atomically precise metalloid nanoclusters (NCs) form a subgroup of this material class[21,22]. The exact knowledge of their structure and composition along with usually smaller sizes, enhanced quantum confinement and the prospect of single-electron switching at room temperature promotes NCs as building blocks for self-assembly[23,24].

Previous studies on Au-NC ensembles have yet either reported conductivity measurements of polycrystalline assemblies[25], along with the first observation of semiconducting properties[26], or the formation of highly ordered microcrystals[20,24,27,28]. However, attempts to quantify the influence of perfect order on the electronic properties of such microcrystals have remained unsuccessful[29]. Overcoming this challenge would allow exploiting the distinct properties of perfectly ordered NC microcrystals, such as superconductance in metalloid $Ga_{84}R_{20}^{4-/3-}$ clusters[18,30,31].

In this paper, we show that assemblies of $Au_{32}(^nBu_3P)_{12}Cl_8$-nanoclusters form idiomorphic microcrystals with high crystallographic phase purity and a strongly preferred growth direction. The crystals are semiconducting and exhibit p-type hopping transport which is limited by Coulomb charging. Energetic disorder is negligible in these microcrystals. In contrast, disordered assemblies of the same clusters show a decrease in the electric conductivity by two orders of magnitude and an over 50% larger activation energy for hopping transport due to the disorder.

## Results

**Self-assembly of $Au_{32}$-NC microcrystals**. The atomically precise building blocks of metalloid $Au_{32}(^nBu_3P)_{12}Cl_8$ nanoclusters (abbreviated as $Au_{32}$-NCs) with an Au-core size of ~0.9 nm are synthesized as previously described[32]. Including the full ligand shell of twelve phosphine ligands and eight chlorine atoms, the building block size is about 1.3 nm, displayed in Fig. 1a. Single-crystal X-ray diffraction of macroscopic crystals of $Au_{32}$-NCs yields a triclinic unit cell containing two crystallographically independent NCs ($a = 1.91$ nm, $b = 1.93$ nm, $c = 3.32$ nm; $\alpha = 73.2°$, $\beta = 86.7°$, $\gamma = 63.4°$, space group $P\bar{1}$)[32].

The preparation process, where dispersed $Au_{32}$-NCs self-assemble into microcrystals at the liquid–air interface and sink into the liquid subphase, is schematically illustrated in Fig. 1b–d. This method allows the preparation of microcrystals onto any

substrate of interest (Further details on the preparation can be found in Methods and Supplementary Information). By 'microcrystals' we understand micrometer-sized idiomorphic single crystals of $Au_{32}$-NCs with high crystallographic phase purity and a strongly preferred growth direction, as detailed below. Figure 1e shows a typical ensemble of self-assembled $Au_{32}$-NC microcrystals with parallelogram shape on a silicon wafer. The crystal shape can be quantified by its geometrical properties of long axis $A$, short axis $B$, angle at the sharp edge $\Delta$, and thickness $h$, as illustrated in the SEM micrograph in Fig. 1f. The lateral expansion (5–30 µm) is 2–3 orders of magnitude larger than the thickness (50–600 nm, see Supplementary Fig. S1), indicating a strongly preferred growth direction. An analysis of SEM micrographs of individual microcrystals yields a distribution of long and short axis, revealing a typical lateral size of $A = 17.4 \pm 4.2$ µm and $B = 10.6 \pm 2.5$ µm, as indicated in Fig. 1g. The lateral size dispersion is calculated to 24%. Further, we observe an aspect ratio of the long and short axis of $A/B = 1.64$ and a sharp edge angle of $\Delta = 63°$ for all microcrystals. This aspect ratio corresponds to the associated ratio found in the unit cell of macroscopic $Au_{32}$-NC crystals, and the angle $\Delta$ suits the $\gamma$-angle of the unit cell of $\gamma = 63.4°$[32]. Hence, the shape of the microcrystals strongly resembles the aforementioned unit cell which renders the crystals idiomorphic. High-resolution SEM images (see Supplementary Fig. S1) reveal perfectly defined edges and extremely flat surfaces, indicating a high crystalline phase purity. Different color impressions in Fig. 1e originate from interference phenomena indicating different thicknesses.

**Structural investigation of self-assembled $Au_{32}$-NC microcrystals**. To verify the crystallinity of self-assembled microcrystals, grazing-incidence small-angle X-ray scattering (GISAXS) measurements are performed, which is a common technique to investigate the structural properties of nanoparticle assemblies in thin films or at interfaces[33–35]. The GISAXS pattern of an ensemble of hundreds of individual microcrystals with different azimuthal orientation (Fig. 1e) is shown in Fig. 2a. Sharp peaks are obtained (Fig. 2c), indicating the high crystallinity of the sample. Doubled peaks in z direction can be observed, caused by a peak splitting phenomenon as previously described[36]. The fit of the obtained peaks yields a triclinic unit cell ($a = 1.9$ nm, $b = 1.94$ nm, $c = 3.48$ nm and $\alpha = 72°$, $\beta = 86°$, $\gamma = 59°$), which is simulated onto the diffraction pattern. The fit is in excellent agreement with the previously determined unit cell of a macroscopic $Au_{32}$-NC single crystal ($a = 1.91$ nm, $b = 1.93$ nm, $c = 3.32$ nm and $\alpha = 73.2°$, $\beta = 86.7°$, $\gamma = 63.4°$)[32]. Considering the GISAXS data together with the morphological appearance of self-assembled $Au_{32}$-NC microcrystals, the unit cell of the microcrystals can be described by a triclinic structure with axis ratios and angles corresponding to a macroscopic single crystal of $Au_{32}$ (Fig. 2d). Thus, microcrystals are µm-sized single crystals, built from individual building blocks of $Au_{32}$-NCs. A typical microcrystal consists of ~5000 unit-cells laterally along the long axis $A$ and ~15–200 unit-cells out-of-plane (~$10^9$ $Au_{32}$-NCs per microcrystal). Furthermore, the dominant first peak in z direction at $q_z \approx 0.37$ Å$^{-1}$ corresponds to a distance of about $d = 1.7$ nm. Assuming this to be the {002}-peak (based on the bulk structure), a unit cell edge of 3.4 nm can be calculated which is in good agreement with the unit cell length $c = 3.32$ nm of the macroscopic NC crystal, indicating that the $c$ axis of the unit cell is aligned along the surface normal. In combination with the missing peaks at {200} and {020}, we conclude that most microcrystals lay flat on the substrate surface, with axis $a$ and $b$ oriented parallel to the substrate, as it is observed by microscopy techniques (Supplementary Fig. S2). Some peaks along the ring-

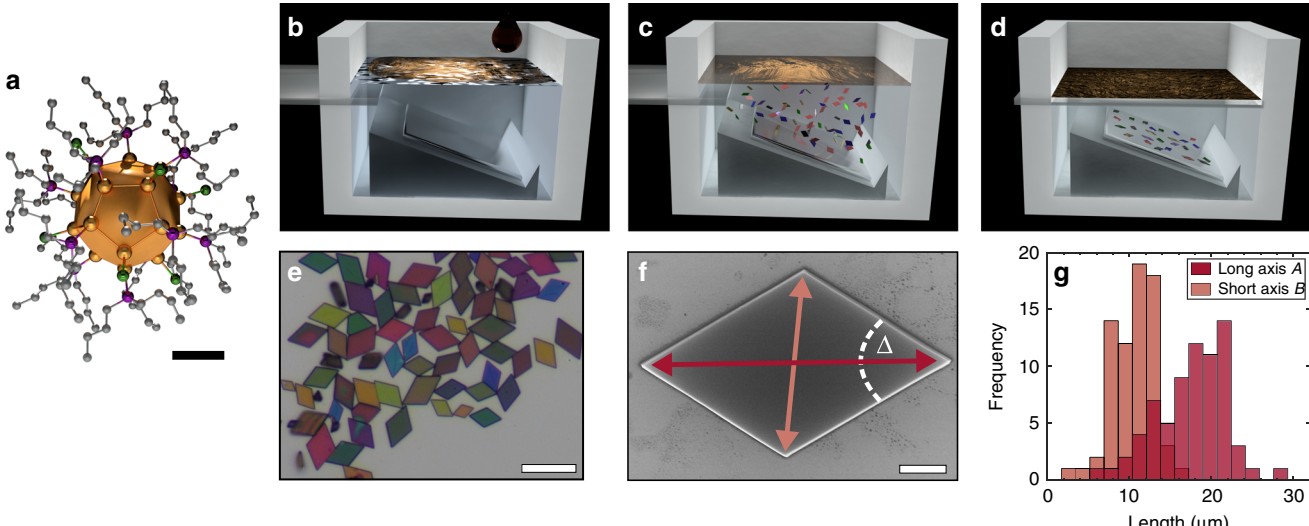

**Fig. 1 Au$_{32}$-NC self-assembly into microcrystals. a** Structural drawing of the Au$_{32}$($^{n}$Bu$_{3}$P)$_{12}$Cl$_{8}$-NCs. The different colors represent the Au- (gold), Cl- (green), P- (purple), and C- (gray) atoms, while hydrogens are omitted for clarity. The Au core has a diameter of ~0.9 nm, while the size of the entire NC is about 1.3 nm. Scale bar: 0.4 nm. **b–d** Schematic illustration of the assembly process. An Au$_{32}$-NC solution is injected onto the liquid subphase within a Teflon chamber. The Au$_{32}$-NCs self-assemble into microcrystals and sink down through the subphase onto the immersed substrate. Details are given in the "Methods" section. **e** Optical micrograph of self-assembled Au$_{32}$-NC microcrystals on a Si/SiO$_{x}$ substrate. The crystals are μm-sized and exhibit a parallelogram shape. Different sizes and thicknesses (color) can be observed. Scale bar: 15 μm. **f** SEM micrograph of a microcrystal with indicated long axis $A$, short axis $B$, and angle $\Delta$. Scale bar: 2 μm. **g** Distribution of long and short axis revealing typical crystal sizes of $A = 17.4 \pm 4.2$ μm, $B = 10.6 \pm 2.5$ μm (dispersity of $Đ = 1.07$, see Supplementary Information for details).

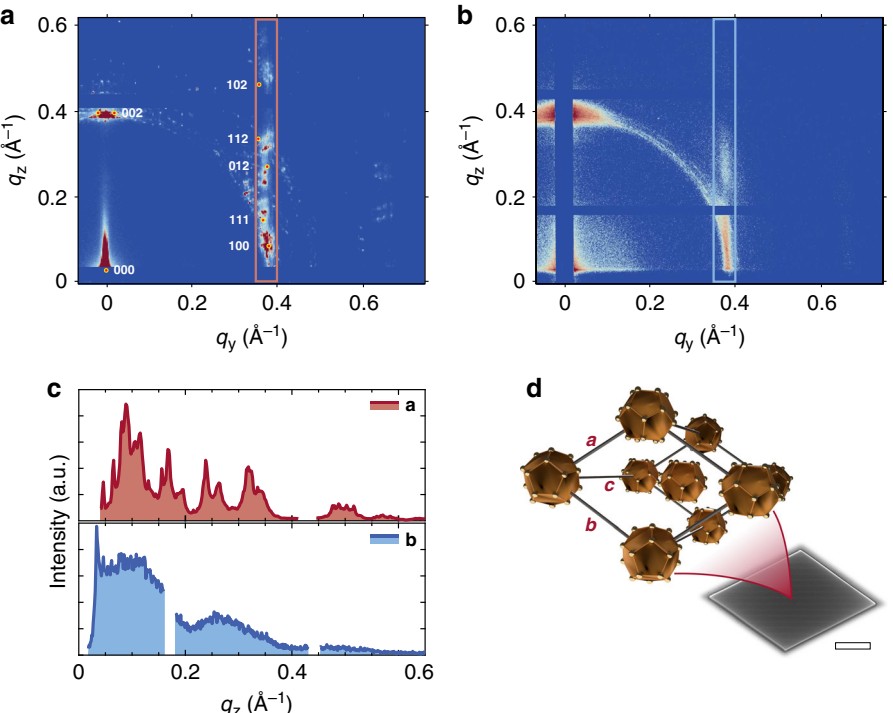

**Fig. 2 Structure of Au$_{32}$-NC microcrystals. a** Grazing-incidence small-angle X-ray scattering (GISAXS) pattern of an ensemble of hundreds of microcrystals with different azimuthal orientation. Diffraction spots are simulated according to a triclinic unit cell ($a = 1.90$ nm, $b = 1.94$ nm, $c = 3.48$ nm, and $\alpha = 72°$, $\beta = 86°$, $\gamma = 59°$). **b** GISAXS pattern of a spin-coated thin film of Au$_{32}$-NCs. Note that the images (a) and (b) in the chosen geometry ($q_{y}$, $q_{z}$) exhibit a very small distortion of the Ewald sphere, which is neglected here. **c** Line scans along $q_{z}$ at $q_{y} = 0.37$ Å$^{-1}$ of the pattern in red (**a**) and blue (**b**), respectively, highlighted by the rectangular boxes. The ensemble of microcrystals (**a**) show distinct sharp peaks, indicating the high crystallinity. The polycrystalline sample (**b**) shows broad signals while lacking clear peaks, indicating the polycrystalline and defect-rich structure. Gaps correspond to detector edges. **d** Schematic drawing of the triclinic unit cell with axis $a$, $b$, and $c$ indicating the idiomorphic growth of the displayed microcrystal. The unit cell contains two crystallographically independent NCs. Ligand spheres are omitted for clarity. The scale bar of the SEM micrograph of a microcrystal corresponds to 3 μm.

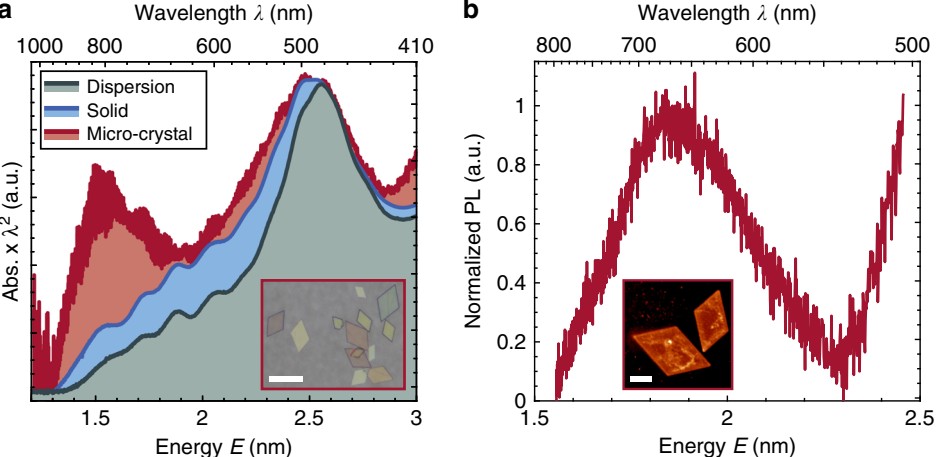

**Fig. 3 Optical properties of Au$_{32}$-NCs and microcrystals. a** Absorbance spectra of Au$_{32}$-NCs dispersed in hexane (green), in a thin film (blue), and in microcrystals (red) on glass. Individual microcrystals show enhanced absorption at 800 nm (1.55 eV), corresponding to the HOMO-LUMO transition. The prominent absorption peak at 481 nm (2.58 eV) for dispersed NCs, is slightly red-shifted to 483 nm (2.51 eV) and broadened for Au$_{32}$-NCs in thin films and microcrystals. All spectra are normalized to the prominent peak at 481 nm. The inset shows an optical micrograph of individual microcrystals on a glass substrate. Scale bar: 15 μm. **b** Photoluminescence (PL) spectrum of an individual microcrystal shows a broad emission peak at around 670 nm. The inset displays the luminescence of two microcrystals upon excitation with 488 nm. Scale bar: 5 μm.

like features at $q \approx 0.37$ Å$^{-1}$ are observed and attributed to single crystals which are not oriented flat on the surface and residual agglomerations which are not Au$_{32}$-NC microcrystals (see Supplementary Fig. S2).

In comparison to microcrystals, the GISAXS pattern of a spin-coated $30 \pm 2$ nm thin film is given in Fig. 2b. Instead of sharp peaks, more ring-like and smeared peaks are observed, clearly indicating the polycrystalline and defect-rich structure of the sample. Throughout this work, we refer to these samples as 'polycrystalline' to indicate their low degree of crystallinity and high angular disorder.

**Optical properties of Au$_{32}$-NC microcrystals**. The comprehensive characterization of the microcrystals is concluded by optical and electronic investigations. Figure 3a displays the energy-corrected absorbance spectra of an Au$_{32}$-NCs dispersion, a thin film, and a microcrystal. Dispersed Au$_{32}$-NCs in solution exhibit several distinct peaks and shoulders, attributed to molecular-like transitions (full spectrum in Supplementary Fig. S3). While the most prominent absorption peak is observed at 2.58 eV (481 nm), the first absorption peak at 1.55 eV (800 nm) corresponds to the HOMO-LUMO transition[37,38]. Most strikingly, only in microcrystals of Au$_{32}$-NCs this peak is strongly enhanced as shown in Fig. 3a (additional spectra of individual microcrystals are given in Supplementary Fig. S4). Further, the absorption onset as well as the most prominent peak at 2.57 eV are red-shifted by ~100 and 10 meV, respectively. A generally enhanced absorption at lower energies and a broadening/shoulder formation at 2.48 eV (500 nm) are observed in microcrystals and thin films of Au$_{32}$-NCs. We attribute these findings to a gradual progression from virtually no electronic coupling between the Au$_{32}$-NCs in solution to weak coupling in thin films and enhanced electronic interactions in the highly ordered microcrystals[2,29,39,40].

While no emission of the Au$_{32}$-NCs is observed in solution, Au$_{32}$-NC microcrystals exhibit photoluminescence resulting in a broad emission peak at 670 nm (1.85 eV) after excitation at $\lambda_{ex} = 488$ nm, as shown in Fig. 3b.

**Electronic properties of Au$_{32}$-NC microcrystals**. To study the possible electronic coupling between individual Au$_{32}$-NCs

observed via optical spectroscopy, we perform (temperature-dependent) conductivity and field-effect transistor (FET) measurements on single Au$_{32}$-NC microcrystals. Most remarkably, we find that the conductivity of highly ordered Au$_{32}$-NCs within microcrystals exceeds that of polycrystalline assemblies by two orders of magnitude, corroborating our hypothesis of enhanced electronic coupling.

We designed electrode devices, in which deposited microcrystals bridge adjacent electrodes to be addressed and probed individually. Details on the device layout are given in the Supplementary Information (Supplementary Figs. S5 and S6). Figure 4a shows an SEM micrograph of a 120-nm-thick microcrystal deposited on two Au electrodes with a gap of $L = 2.8$ μm on a Si/SiO$_x$ device. Figure 4b displays a typical $I–V$ curve of an individual microcrystal in the range of $\pm 200$ mV. Ohmic behavior (at room temperature) in the low-field regime (up to $\pm 1$ V) is observed. Electrical conductivity values with typical uncertainties of <10% are calculated from these measurements for 54 individual microcrystal channels on different devices. Figure 4c displays the narrow distribution of conductivity values, showing a mean conductivity of $\sigma = 1.56 \times 10^{-4}$ S/m with a standard deviation of $\pm 0.90 \times 10^{-4}$ S/m. In contrast, the mean conductivity of polycrystalline thin films of Au$_{32}$-NCs is only $\sigma \approx 1 \times 10^{-6}$ S/m (Supplementary Fig. S7). These devices are obtained by spin-coating on substrates with interdigitated electrodes of channel length $L = 2.5$ μm and width $W = 1$ cm. The film thicknesses are in the range of $30 \pm 2$ nm to $47 \pm 4$ nm (Supplementary Figs. S8 and S9).

To shed light on the charge-transfer mechanism of electronic transport within microcrystals and polycrystalline films of Au$_{32}$-NCs, we perform temperature-dependent conductivity measurements at $T = 340–170$ K (Fig. 4d). Below this range, the measured current approaches the noise level. The measured temperature dependence can be described by an Arrhenius-type temperature-activated hopping (Supplementary Fig. S10)[41]. Fitting the conductivity data accordingly, we obtain activation energies of $E_A = 227 \pm 17$ meV for individual microcrystals and $E_A = 366 \pm 62$ meV for the spin-coated polycrystalline Au$_{32}$-NCs thin films.

To further characterize the electronic properties of self-assembled Au$_{32}$-NC microcrystals and polycrystalline films, FET measurements are performed. Strikingly, a field-effect can

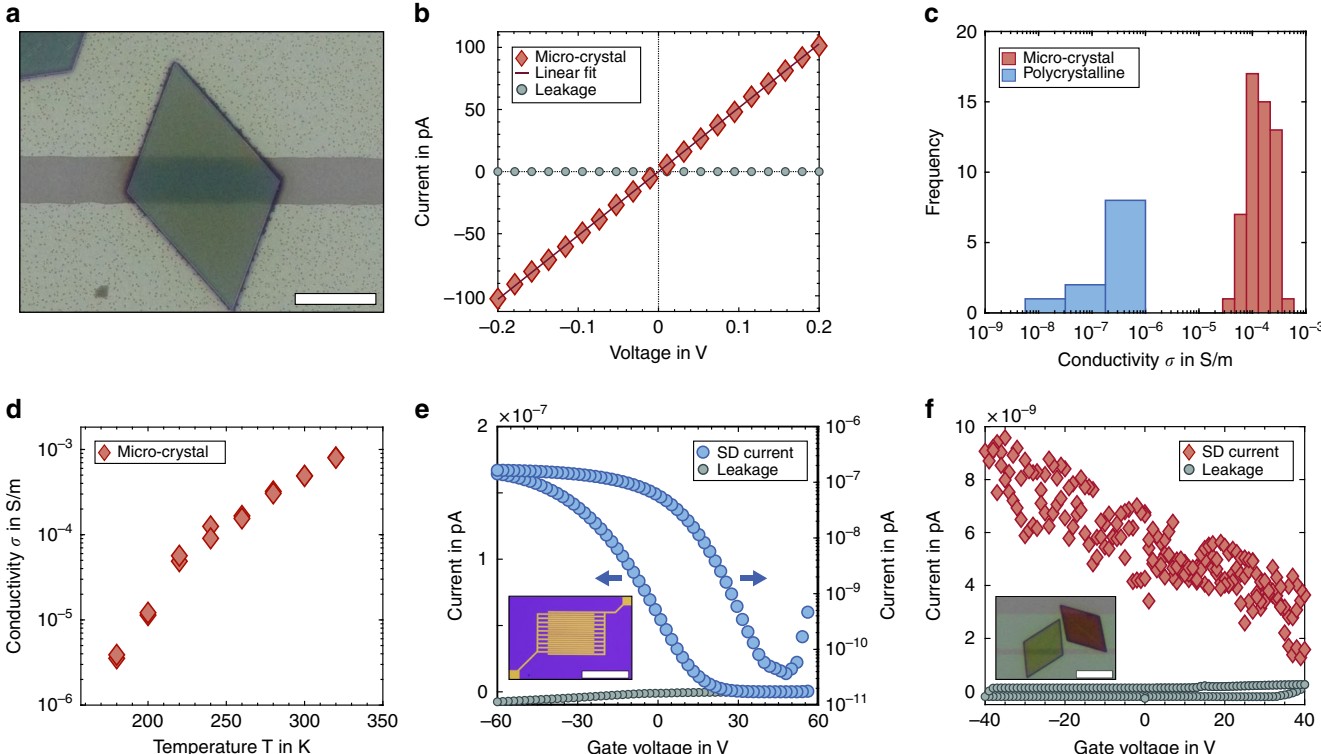

**Fig. 4 Electronic properties of Au$_{32}$-NC microcrystals. a** SEM micrograph of an individual microcrystal deposited on two horizontal Au electrodes on a Si/SiO$_x$ device. The electrodes form a channel with length $L = 2.8\,\mu m$. The width and height of the contacted microcrystal are $W = 7.9 \pm 0.4\,\mu m$ and $h = 120\,nm$. SEM and optical micrographs are merged. Scale bar: 5 $\mu m$. **b** Typical I–V curve of an individually probed microcrystal. Ohmic behavior is observed in the low voltage regime. **c** Distribution of conductivity $\sigma$ of 54 individual microcrystals and 19 polycrystalline thin films. The conductivity of microcrystals exceeds that of polycrystalline films by ~2 orders of magnitude. **d** Temperature-dependent conductivity of Au$_{32}$-NC microcrystals with two individual measurements per temperature step. **e** FET transfer curve (blue) of a polycrystalline film of Au$_{32}$-NCs on an interdigitated electrode device with $L = 2.5\,\mu m$, $W = 1\,cm$, measured at $V_{SD} = 10\,V$ on a linear and logarithmic scale together with the negligible leak current (gray). Arrows indicate the corresponding y-axis. **f** FET transfer curve (red) of an individual microcrystal device with $L = 1.5\,\mu m$, $W = 10.4 \pm 0.2\,\mu m$, measured at $V_{SD} = 5\,V$, together with the negligible leak current (gray). The insets in (**e**) and (**f**) display optical micrographs of the two devices. Scale bars correspond to 500 and 10 $\mu m$, respectively.

be observed, indicating semiconducting behavior of the metal NC assemblies. The tri-butyl-phosphine ligands covering the cluster cores limit the electronic coupling enough to prevent metallic behavior[26]. Figure 4e shows the FET transfer curve of a polycrystalline thin film of Au$_{32}$-NCs on interdigitated electrodes with channel dimensions of $L = 2.5\,\mu m$, $W = 1\,cm$ and $h = 30 \pm 2\,nm$. p-type behavior is observed, indicating holes (h$^+$) as majority charge carriers. The current flow can be modulated by more than three orders of magnitude (ON/OFF ratio of ~4000). Ambipolar behavior is also observed for very high threshold voltages of $V_G > 40\,V$. The calculated hole mobility of spin-coated Au$_{32}$-NC films is in the range of $\mu(h^+) \sim 10^{-6} – 10^{-5}\,cm^2\,V^{-1}\,s^{-1}$.

Figure 4f displays the FET transfer curve of an individual Au$_{32}$-NC microcrystal, which also indicates p-type behavior. Note that the current flows through a much smaller channel width $W$ of 5–10 $\mu m$ in this case. Here, the mean value and standard deviation of the hole mobility of individual microcrystals can be calculated to be $\mu(h^+) = 0.8 \times 10^{-4} \pm 0.58 \times 10^{-4}\,cm^2\,V^{-1}\,s^{-1}$. Values up to $2 \times 10^{-4}\,cm^2\,V^{-1}\,s^{-1}$ are observed (Supplementary Fig. S11). The noise in the current flow and the low modulation can be attributed to the non-ideal channel geometry. Further, the quality of contact between the dielectric SiO$_x$ layer and the microcrystal is not known. The non-ideal contact might influence the appearance of transfer curves (details are given in Supplementary Fig. S12). We have verified that the contact resistance of Au$_{32}$-NC microcrystal and thin-film devices is negligible (Supplementary Fig. S13).

Knowing the charge carrier mobility $\mu(h^+)$ and the conductivity of individual Au$_{32}$-NC crystals, we calculate the charge carrier concentration to be $n(h^+) = 2 \times 10^{17}\,cm^{-3}$. This corresponds to one free charge carrier per 1000 Au$_{32}$-NCs, as the concentration of individual Au$_{32}$-NC within a crystal is $1.9 \times 10^{20}\,cm^{-3}$.

## Discussion

The Au$_{32}$-NC HOMO-LUMO gap of 1.55 eV (Fig. 3a) is consistent with earlier reports on other Au NCs and the expected degree of quantum confinement. Specifically, for NCs with 11 and 25 Au atoms and, thus, stronger quantum confinement, HOMO-LUMO transitions of 2.97 and 1.84 eV have been reported[26,42]. In line with this, (AuAg)$_{34}$-NCs exhibit a HOMO-LUMO transition of around 1.4 eV[20]. A related size-dependent study of NCs with 10–39 Au-core atoms revealed HOMO-LUMO transitions from 3.7 to 1.7 eV[43].

The solid-state luminescence of the Au$_{32}$-NCs (Fig. 3b) at 1.85 eV is fully consistent with the emission of other Au NCs[21,43–46], such as Au$_{25}$[37,47–49], and may be attributed to aggregation-induced emission[48,50,51]. In contrast to the HOMO-LUMO transition, which is believed to involve a (mostly dark) sp-intraband transition, the luminescence in Au$_{25}$ and Au$_{28}$-NCs results from an sp→d interband transition, which may also be the case in Au$_{32}$[46,47]. We note, however, that Au$_{25}$-NCs consist of an icosahedral Au$_{13}$ core, while the core of the Au$_{32}$-NC is a hollow Au$_{12}$ icosahedron with potentially different optical properties[21,32,52].

The conductivity (Fig. 4c) and mobility (Fig. 4e) of the thin polycrystalline $Au_{32}$-NC films are in good agreement with previously reported values for $Au_{25}$- and $Au_{38}$-NCs[26,53]. In contrast to the study by Galchenko et al. on $Au_{25}$-NCs with n-type transport[26], we observe here p-type behavior or ambipolar transport with extremely high threshold voltages of $\sim V_G = +50$ V. In a recent study by Yuan et al.[20], single crystals of $(AuAg)_{34}$-NCs also exhibit p-type behavior with mobilities of $\sim 2 \times 10^{-4}$ $cm^2 V^{-1} s^{-1}$ and an ON/OFF ratio of $\sim 4000$. The conductivity of $Au_{32}$-NC microcrystals exceeds that of monomeric and polymerized $(AuAg)_{34}$-NC crystals ($6 \times 10^{-8}$ S/m and $1.5 \times 10^{-5}$ S/m, respectively)[20].

The key finding of this work is that the above-mentioned properties change dramatically as long-range order is introduced to the NC ensembles (Fig. 2a). While the seminal work by Li et al.[28] reported electric transport measurements on similar single crystals for the first time, we provide a direct comparison of the transport properties in the ordered vs. the glassy state. This uniquely allows us to quantify the value of long-range order for electric transport in Au-NC ensembles.

To this end, we use the experimentally determined activation energies to charge transport in the $Au_{32}$-NCs, either as microcrystals ($E_A = 227 \pm 17$ meV) or as polycrystalline thin films ($E_A = 366 \pm 62$ meV). Transport in weakly coupled nanostructures depends on the transfer integral ($\delta$), the Coulomb charging energy ($E_C$), and the energetic disorder $\Delta\alpha$[54]. Strongly temperature-activated transport (Fig. 4d) suggests that even the $Au_{32}$-NC microcrystals are in the Mott regime with $E_C \gg \delta$. Thus, charge transport is dominated by $E_C$ and possibly $\Delta\alpha$. $E_C$ can be referred to as the self-capacitance of the NC and it describes the required energy for addition or removal of an additional charge carrier to the NC. We estimate $E_C$ of the microcrystals to 276 meV (for details, see Supplementary Information), which is consistent with the full activation energy. Thus, charge carrier transport in the microcrystals depends solely on the charging energy and the energetic disorder is negligible. In contrast, $E_A$ in the polycrystalline thin films largely exceeds $E_C$, suggesting a significant degree of energetic disorder, which is caused by structural, orientational, or chemical disorder of the individual NCs. Since the NCs are atomically precise, we hold only structural defects, such as grain boundaries, cracks, and a lack of orientational order to be responsible for the occurrence of a nonzero $\Delta\alpha$ in the polycrystalline films[55]. This effect is especially pronounced here, as systems with large $E_C$ are generally very sensitive towards structural disorder[27]. In contrast, $Au_{32}$-NC microcrystals not only consist of chemically identical building blocks but also exhibit structural perfection, which manifests in a vanishing value of $\Delta\alpha$. We suggest that this is the reason for the enhanced electronic coupling and altered optoelectronic properties. Future attempts to further increase coupling in Au-NC microcrystals should focus on increasing the transfer integral, for instance by reducing the distance between adjacent clusters or by covalent coupling with conjugated linkers. If $\delta \approx E_C$, a Mott insulator-metal transition occurs and band-like transport becomes possible. The basis for this will be atomically defined building blocks in combination with a suitable coupling as pioneered here.

In conclusion, atomically precise $Au_{32}(^nBu_3P)_{12}Cl_8$ nanoclusters are self-assembled into microcrystals with high crystallographic phase purity and a strongly preferred growth direction. Individual microcrystals exhibit semiconducting p-type behavior and temperature-activated hopping transport, limited by Coulomb charging. Most strikingly, additional optical transitions emerge, and charge carrier transport is enhanced by two orders of magnitude in the microcrystals compared to polycrystalline thin films, highlighting the advantageous effect of long-range structural order. This study implies that utilizing atomically precise building blocks for the self-assembly into superlattices eliminates energetic disorder and provides a promising route towards self-assembled nanostructures with emergent optoelectronic properties.

## Methods

**Materials**. All chemicals were used as received unless otherwise noted. Octane, ethanol, dichloromethane, and acetonitrile were bought from Sigma-Aldrich and were degassed and distilled before usage. $NaBH_4$ was bought from Acros Organics. Silicon/silicon dioxide $(Si/SiO_x)$ wafer with 200 nm $SiO_x$ layer and n-doped Si were purchased from Siegert Wafer. Photoresist, developer, and remover (ma-N 405, ma-D 331/S, and mr-Rem660, respectively) were purchased from micro resist technology GmbH, Berlin.

**Synthesis of $Au_{32}(^nBu_3P)_{12}Cl_8$-nanoclusters**. 1 mmol of $^nBu_3PAuCl$ was dissolved in 20 ml of ethanol before a suspension of 38 mg of $NaBH_4$ in ethanol was added. The reaction solution was stirred for 1 h before the solvent was removed under reduced pressure. The residual black solid was extracted with $CH_2Cl_2$ and layered with three times the amount of diethyl ether. After 1 week a gold mirror formed while a dark supernatant remained. The dark brown supernatant was filtered off and concentrated under vacuum. Crystals of $Au_{32}(^nBu_3P)_{12}Cl_8$ formed by storing the solution at $-30\,°C$ for a few days.

**Self-assembly of $Au_{32}$-NC microcrystals**. The formation of crystals via liquid–air interface method is schematically illustrated in Fig. 1b–d. A solution of $Au_{32}$-NCs in octane (200 μl, 0.5 mM) was added onto a subphase of acetonitrile inside a home-built Teflon chamber (Fig. 1b). The self-assembly of $Au_{32}$-NCs into microcrystals took place at the phase boundary between the acetonitrile subphase and the NC solution upon evaporation of the solvent. The microcrystals started to sink down through the subphase and stuck to the desired substrate which was previously placed inside the liquid subphase (Fig. 1c). After 45 min a glass slide was horizontally inserted into the subphase to separate the residual $Au_{32}$-NC membrane (floating on the liquid–air interface) from the bottom substrate (Fig. 1d). The liquid subphase was removed and the substrate dried at ambient conditions. Microcrystal fabrication took place at ambient condition. Further details are given in the Supporting Information and Fig. S14.

**Microcrystal device fabrication**. For the microcrystal electrode devices, standard photolithography technique (negative tone resist) was used to pattern Au electrodes on $Si/SiO_x$ substrates (200 nm $SiO_x$). Au (8–10 nm) and Ti (~2.5 nm) as an adhesion layer were thermally evaporated under high vacuum conditions. Ultrasonic-assisted lift-off in mr-Rem660 removed the residual resist and metal layer. Electrodes with gaps of 1.5–2.5 μm (channel length $L$) were realized. Devices were coated with microcrystals as described above and checked with a basic light microscope to identify channels, where a single microcrystal bridges two adjacent electrodes.

**Thin-film fabrication**. Thin-film samples for absorbance, GISAXS, and electronic measurements were prepared as follows. For thin-film electronic devices with interdigitated electrodes, commercially available OFET substrates (Fraunhofer IPMS, Dresden) were purchased. For GISAXS and absorbance measurements, Si wafer with 200 nm $SiO_x$ layer and glass slides were used, respectively. The substrates ($15 \times 15$ mm$^2$) were coated with 100 μl of a 0.5 mM $Au_{32}$-NC solution (hexane or heptane) and spin-coated after 2 min with a speed of 760 rpm or 2000 rpm for 30 s. All devices were prepared at ambient conditions in a fume-hood. The thickness of thin films was determined by profilometry (Dektak XT-A, Bruker), details are given in the Supporting Information.

**Grazing-incidence small-angle X-ray scattering**. GISAXS measurements were conducted on a Xeuss 2.0 setup (Xenocs). A CuK$\alpha$ X-ray beam with wavelength $\lambda = 1.5418$ Å ($E = 8.04$ keV) and a beam size of ~500 × 500 μm$^2$ (FWHM) was used. A two-dimensional detector Pilatus 300 K (Dectris) with 487 × 619 pixels of 175 × 175 μm$^2$ was positioned 365 mm downstream of the sample. The samples (microcrystal ensemble or thin film on Si wafer with 200 nm $SiO_x$ layer, both described as above) were probed at an incidence angle of 0.2°. Acquisition times of 60 and 30 min were used to obtain the GISAXS maps in Fig. 2a, b, respectively. Simulated peaks correspond to a triclinic unit cell with $a = 1.90$ nm, $b = 1.94$ nm, $c = 3.48$ nm and $\alpha = 72°$, $\beta = 86°$, $\gamma = 59°$, which is in good agreement with the X-ray diffraction data from macroscopic $Au_{32}$ NC crystals ($a = 1.91$ nm, $b = 1.93$ nm, $c = 3.32$ nm and $\alpha = 73.2°$, $\beta = 86.7°$, $\gamma = 63.4°$). Simulations were performed using the MATLAB toolbox GIXSGUI[56].

**Optical measurements**. Absorbance spectra of $Au_{32}$-NC in solutions (0.5 mM in hexane) were acquired with an UV-vis-NIR spectrometer (Cary 5000, Agilent Technologies). For thin films spin-coated on glass slides (as described above), a Perkin Elmer Lambda 950 spectrometer was used. For individual microcrystals on

glass slides, an inverted microscope (Nikon Eclipse Ti-S) with a spectrometer was used. The sample was illuminated with unpolarized white light by a 100 W halogen lamp. The transmitted light was collected by a ×60 objective (Nikon, CFI S Plan Fluor ELWD, NA = 0.7). The collected light was passed to a grating spectrograph (Andor Technology, Shamrock SR-303i) and detected with a camera (Andor Technology, iDusCCD). All absorbance spectra were energy-corrected using the expression $I(E) = I(\lambda) \times \lambda^2$ [37,57]. Photoluminescence images and emission spectra of individual $Au_{32}$-NC microcrystals were acquired with a home-built confocal laser scanning microscope. The diode laser (iBeam smart, Toptica Photonics) was operated in continuous wave Gaussian mode at an excitation wavelength of $\lambda_{ex} = 488$ nm. Luminescence images were obtained with a photon-counting module (SPCM-AQR-14, Perkin Elmer) and spectra were acquired with an UV-VIS spectrometer (Acton SpectraPro 2300, Princeton Instruments). The background was subsequently subtracted from the emission spectra.

**Scanning electron microscopy**. SEM imaging of microcrystals on $Si/SiO_x$ devices was performed with a HITACHI model SU 8030 at 30 kV. To estimate the thickness of microcrystals, samples were titled by 85° with respect to the incoming electron beam.

**Electrical measurements**. All electrical measurements were conducted under vacuum in a probe station (Lake Shore, CRX-6.5 K). All samples were placed under vacuum overnight before measurement (pressure of $<10^{-5}$ mbar). Au-electrode pairs were contacted with W-tips, connected to a source-meter-unit (Keithley, 2636 B). A back electrode worked as gate electrode. For two-point conductivity measurements, voltage sweeps in a certain range of ±1 V were applied and the current (as well as leak current) detected. Fitting the linear I–V curve yielded the conductance value G. Conductivity $\sigma$ was calculated as $\sigma = (G \times L)/(W \times h)$. The dimensions length, width, thickness $(L, W, h)$ were determined by SEM imaging for microcrystals or profilometry for spin-coated thin films (Dektak XT-A, Bruker). For FET measurements (bottom-gate, bottom-contact configuration), a source-drain voltage of $V_{SD}$ was applied and $I_{SD}$ was measured, modulated by applied gate voltages $V_G$. Using the gradual channel approximation, field-effect mobilities $\mu$ were calculated (Supplementary Equation S2).

For temperature-dependent measurements, the devices were cooled down to 8 K and gradually heated with a Lake Shore temperature controller (model 336). Current was detected in the temperature range 170–340 K. At least two measurements were taken for every temperature. After reaching 340 K, measurements were repeated at lower temperature, to verify the reversibility. The temperature-activated hopping behavior can be described as an Arrhenius-type, which is expressed in Eq. (1)[41].

$$\sigma = \sigma_0 \exp(-E_A/k_B T) \qquad (1)$$

Here, $E_A$ is the activation energy, $k_B$ the Boltzmann constant, $T$ the temperature, and $\sigma_0$ a constant. $E_A$ was obtained from the slope of $\ln(\sigma)$ as a function of $T^{-1}$.

## Data availability

Single-crystal X-ray data of the nanocluster $Au_{32}(^nBu_3P)_{12}Cl_8$ is available free of charge at www.ccdc.cam.ac.uk/conts/retrieving.html or from Cambridge Crystallographic Data Center (CCDC number 1883285). The data that support the findings of this study are available from the corresponding author upon reasonable request.

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

## Acknowledgements
F.F. thanks the PhD Network: "Novel nanoparticles: from synthesis to biological applications" at the University of Tübingen for financial support. This project has been funded by the Carl Zeiss Stiftung (Forschungsstrukturkonzept "Interdisziplinäres nanoBCP-Lab") as well as the European Research Council (ERC) under the European Union's Horizon 2020 research and innovation program (grant agreement No 802822). M.H. thanks the Alexander von Humboldt Foundation for financial support. SEM measure-ments using a Hitachi SU 8030 SEM were funded by the DFG under contract INST 37/829-1 FUGG. Support by C. Dreser, S. Dickreuter, and A. Bräuer with the optical setup of the M. Fleischer group is gratefully acknowledged. We thank B. Fischer for his support during the interference reflection microscopy measurements. Open Access funding enabled and organized by Projekt DEAL.

## Author contributions
F.F. synthesized the NCs and developed the microcrystals fabrication. A.M. performed the device fabrication, SEM measurements, and analyzed the electrical measurements. F.F. and A.M. conducted optical absorbance measurements, electrical measurements, interpreted the results, and wrote the manuscript. M.H. performed the GISAXS measurements and analysis. O.G., K.B., and A.J.M. conducted the luminescence measurements. F.S., A.S., and M.S. conceived and supervised the project. All authors have given approval to the final version of the manuscript.

## Funding

## Competing interests
The authors declare no competing interests.
