## [Peer Review File · Nature Communications]

REVIEWER COMMENTS

Reviewer #1 (Remarks to the Author):

This manuscript reports the self-assembly of atomically precise gold nanoclusters into micro-crystals. These crystals were characterized in details by single-crystal X-ray diffraction, GISAXS, and SEM. Through further study on the optical and electronic properties, it was revealed that the long-range structural order of such a self-assembly system led to stronger electronic couplings in the solid state, higher conductivity, and a lower energy barrier for hopping, compared to the polycrystalline thin film made of the same gold nanoclusters. Overall, this paper represents an in-depth study on a crystalline self-assembly of atomically precise gold nanoclusters, and provides solid evidence to elucidate the impacts and stronger electronic couplings brought by the long-range order in crystals. I find this manuscript interesting to readers from different fields including inorganic chemistry, supramolecular assembly, material science and engineering considering the broad readership of Nature Communications. However, there are a few minor concerns that the authors should address while revising this manuscript.

1. The authors attributed such a strong coupling to the high ordering in the crystalline lattice of the self-assembled micro-crystals. In addition to this, the short distance between the gold nanocluster cores, as illustrated in the crystal structures (in Figure 2d), should also be considered. The butyl groups on the surface of the gold core prevent metallic contact but still contribute to the strong coupling. The authors should comment on this point in their revised manuscript.
2. On Page 3 Line 77, it is worthy to note that “polydispersity” has been deprecated by IUPAC.
3. Some data regarding the measurement of thin film samples are missing in the manuscript and supporting information, including a presentative I-V curve in the conductivity measurement, and temperature-dependent conductivity diagram. Considering these are very important control experiments in this work, the authors should add the original data in the manuscript or supporting information, instead of only showing the numbers.
4. Please add in the Methods section the details on preparation of thin-film samples for GISAXS, and absorbance measurements, and the method used for thin film thickness measurements. Also, since the authors fabricated these thin films by spin coating, hav these films been treated with any post-coating techniques such as annealing to remove the solvents?
5. Please add the axis name to the right side of Figure 4e.
6. Given the very low current intensity and high noise in Figure 4f, this data should be evaluated more carefully. The authors attributed the noise to “non-ideal channel geometry”, which is not that convincing to me, because the channel size is only needed for the post-measurement calculation of mobility, and should not affect the I-V curve that much during the measurement. Also, the authors showed the mean values of mobility and should also include the standard derivation.
7. In Abstract, the meaning of “structure-transport” is vague and confusing. In Line 22, it should “resulted from” not “resulting from”.
8. In Line 60 Page 3, “100-fold decrease” is not a correct expression. “1-fold decrease” meaning going from 100% to 0%. Here, it should be “decrease to 1%” if that’s what the authors were trying to say.
9. Authors should make a broader intellectual framing for their work since the “transport” issue in the

hierarchical materials is not limited to gold nanoclusters. Other cluster-based materials should be mentioned in the introduction. Some notable recent examples: Nature Mater. 2017, 16, 83-88, Nature Mater. 2017, 16, 474-480, Nature Mater. 2018, 17, 341-348.

Reviewer #2 (Remarks to the Author):

In this article, Fetzner et al reported the electrical conductivity and charge transport behavior of the Au₃₂(Bu₃P)₁₂Cl₈ micro-crystals and films. The micro-crystals were formed by self-assembly at the liquid-air interface. A key conclusion is that the electrical conductivity of Au₃₂ micro-crystals is 100-fold higher than the Au₃₂ film (10⁻⁴ vs 10⁻⁶ S/m). Temperature-dependent conductivity measurements show that hopping activation energy is lower in micro-crystal than in the film (227 meV vs 366 meV), indicating the film is more disordered. The manuscript was well organized and the data was carefully presented. However, there are some major issues need to be further adjusted.

The authors suggested that the increased conductivity is due to the enhanced electronic coupling in micro-crystals, as highlighted in the article title. However, the lower conductivity of thin film could just be a result of cracks or inhomogeneity in the large area of spin-coated films, as suggested in Line 306-308. Lacking disorder in micro-crystal does not necessarily mean there is an electronic coupling between clusters. Also, there is no direct evidence or theoretical demonstration of enhanced electronic coupling such as the formation of extended band structures. Please consider adjusting the title to better reflect the key findings such as “enhanced electrical conductivity/charge transport...” instead of using the vaguely defined “electronic coupling”.

Compared with characterizations on the micro-crystals, there is little information about the film quality, such as smoothness and uniformity. These are also critical data since the conductivity of the thin film is significantly affected by the quality of the film. The authors mentioned that the thickness of the film is only 30-50 nm, which is much thinner than the thickness of the micro-crystals (50-600 nm). The thinner film could have more defects such as pinholes. If the quality or the thickness of the film is increased, will the conductivity be increased accordingly?

For the absorption spectrum of micro-crystals, there is an additional very broad peak at 1.55 eV, which the author assigned as the HOMO-LUMO transition (line 162, 176-178). However, the additional peak can be a result of interference phenomena since the thickness of the crystal is in the same regime of Vis-NIR wavelength (line 107). Can the author provide any evidence to exclude this possibility?

Line 290-292, it is not proper and meaningful to claim the first time. For example, in ref 18, the conductivity was measured on a single crystal. Consider remove or rephrase the sentence.

Typo: line 252, mobility should be 10⁻⁶ to 10⁻⁵.

Reviewer #3 (Remarks to the Author):

In this manuscript, the authors reported the controlled self-assembly of atomically precise Au₃₂(nBu₃P)₁₂Cl₈ nanoclusters into micro-crystals. By comparing micro-crystal with dispersion and thin film, they demonstrated that the long-range order in micro-crystal enhances electronic coupling, resulting in significant enhancement of electric conductivity and charge carrier mobility. Generally, the data and analysis are comprehensive and solid. However, as despite the significant work done, I think its current version does not belong to Nature Communication, unless the authors can provide more solid measurement data to support their conclusion.

Below are a few comments:

1. The crystal structures of micro-crystals and structures of thin films should be characterized clearly by X-ray single crystal diffractometer to determine how Au₃₂-NCs interact with neighboring clusters, which is critical for explaining why the electronic coupling is enhanced in micro-crystals relative to in thin films.
2. How the Au₃₂-NCs assemble into micro-crystals? And what is the structure difference between single crystals and thin films. This could help to explain the claim that the electronic coupling of micro-crystals is enhanced comparing with thin films.
2. The claims that "this first structure-transport correlation on self-assembled superstructures of atomically precise gold nanoclusters" and "however, attempts to quantify the influence of perfect order on the electronic properties of such micro-crystals have remained unsuccessful" are not objective enough since reference 18 has already reported the structure-transport correlation of Au NPs single crystal.
3. Please try to correlate the semiconductor characteristic and hole-conduction mechanism with the unique structure of the micro-crystals.
4. Since the micro-crystals has strongly preferred growth direction, it is better to compare the electrical properties of different crystallographic axes.
5. The conductivity of micro-crystal is tested with two-point probe method. It will be better to conduct a 4-point probe measurement for this system. This is recommended for measuring electrical conductivity in metal-organic framework materials. See: J. Am. Chem. Soc. 2016, 138, 44, 14772-14782
6. Figure 1b-d and Figure 4a should be marked clearly.

Reviewer comments are denoted in black.

Author replies are denoted in green.

Specific changes in the manuscript are denoted in red. Indicated line numbers correspond to the revised manuscript.

REVIEWER COMMENTS

Reviewer #1 (Remarks to the Author):

This manuscript reports the self-assembly of atomically precise gold nanoclusters into micro-crystals. These crystals were characterized in details by single-crystal X-ray diffraction, GISAXS, and SEM. Through further study on the optical and electronic properties, it was revealed that the long-range structural order of such a self-assembly system led to stronger electronic couplings in the solid state, higher conductivity, and a lower energy barrier for hopping, compared to the polycrystalline thin film made of the same gold nanoclusters. Overall, this paper represents an in-depth study on a crystalline self-assembly of atomically precise gold nanoclusters, and provides solid evidence to elucidate the impacts and stronger electronic couplings brought by the long-range order in crystals. I find this manuscript interesting to readers from different fields including inorganic chemistry, supramolecular assembly, material science and engineering considering the broad readership of Nature Communications. However, there are a few minor concerns that the authors should address while revising this manuscript.

1. The authors attributed such a strong coupling to the high ordering in the crystalline lattice of the self-assembled micro-crystals. In addition to this, the short distance between the gold nanocluster cores, as illustrated in the crystal structures (in Figure 2d), should also be considered. The butyl groups on the surface of the gold core prevent metallic contact but still contribute to the strong coupling. The authors should comment on this point in their revised manuscript.

Thank you for this remark. We comment on this point in the revised manuscript: “The tri-butyl-phosphine ligands covering the cluster cores limit the electronic coupling enough to prevent metallic behavior [Adv. Mater. 2019, 31, 190068]” (line 266–268).

2. On Page 3 Line 77, it is worthy to note that “polydispersity” has been deprecated by IUPAC.

We thank the reviewer for this remark. We changed the term “polydispersity” to “dispersity” and calculated the value according to the IUPAC definition. The correct value of $D=1.07$ is given in the revised manuscript (line 96) and details are given in the revised SI (line 670–678).

3. Some data regarding the measurement of thin film samples are missing in the manuscript and supporting information, including a presentative I-V curve in the conductivity measurement, and temperature-dependent conductivity diagram. Considering these are very important control

experiments in this work, the authors should add the original data in the manuscript or supporting information, instead of only showing the numbers.

Thank you for this suggestion. A representative I-V curve for spin-coated thin films together with a direct comparison with the I-V curve of a self-assembled micro-crystal have been included in the revised SI (figure S7, line 740–752). A typical temperature-dependent conductivity diagram has also been included in the revised SI (figure S10, line 800–818).

Figure S7: *I-V* curves of Au₃₂-NC devices. **(a)** Typical *I-V* curve of an individually probed micro-crystal. Fitting I_{SD} (red) yields the conductance $G_{\text{crystal}} = 57$ pS. The leakage (blue) is negligible. The inset displays the corresponding micro-crystal with $L = 2.8$ μm , $W = 6.8$ μm and $h = 98$ nm. The conductivity can be calculated to $\sigma_{\text{crystal}} = 2.4 \times 10^{-4}$ S/m. Scale bar: 7 μm . **(b)** Typical *I-V* curve of Au₃₂-NC thin film. Fitting I_{SD} (red) yields the conductance $G_{\text{film}} = 533$ pS. The leakage (blue) is negligible. The inset displays the corresponding device with $L = 2.5$ μm , $W = 1$ cm and $h = 30 \pm 2$ nm. The conductivity can be calculated to $\sigma_{\text{film}} = 4.4 \times 10^{-6}$ S/m. Scale bar: 250 μm .

Figure S10: Temperature-dependent conductivity measurements of an Au₃₂-NC micro-crystal (a,b) and a spin-coated thin film (c,d). **(a)** Conductivity as a function of temperature of an individual micro-crystal with $L = 2.8$ μm , $W = 7.9 \pm 04$ μm and $h = 120$ nm. At every temperature step, several measurements were performed. This Figure corresponds to Figure 4d. **(b)** Arrhenius-plot of the data points shown in (a). The linear fit yields the activation energy $E_A \approx 0.2$ eV. The R^2 value of 0.99 indicates the goodness of the linear fitting. **(c)** Conductivity as a function of temperature of a thin film channel with $L = 2.5$ μm , $W = 1$ cm and $h = 30 \pm 2$ nm. At every temperature step, several measurements were performed. **(d)** Corresponding Arrhenius-plot of (c). The linear fit yields the activation energy $E_A \approx 0.33$ eV. The R^2 value of 0.92 indicates the goodness of the linear fitting. In the manuscript, we report the mean value of E_A for micro-crystals and thin films, including the standard deviation, based on the evaluation of 25 and 7 of such plots, respectively. This yields $E_A = 227 \pm 17$ meV for micro-crystals and $E_A = 366 \pm 62$ meV for the thin films.

4. Please add in the Methods section the details on preparation of thin-film samples for GISAXS, and absorbance measurements, and the method used for thin film thickness measurements. Also, since the authors fabricated these thin films by spin coating, have these films been treated with any post-coating techniques such as annealing to remove the solvents?

The details on the preparation of thin film samples for absorbance and GISAXS measurements, as well as the film thickness characterization have been added to the methods section of the revised manuscript (line 535–543). Additional SEM images and information on the thickness characterization are given in the revised SI (Figure S8, S9, line 764–794, line 283–284).

All thin film samples were prepared by spin coating and no post-coating techniques were applied. The samples were stored over night to ensure full evaporation of residual solvents. Thin film samples for electronic measurements were placed under vacuum overnight in the probe station and measured under vacuum conditions (pressure of $\leq 10^{-5}$ mbar).

This information has also been added in the revised SI (line 764-769).

Figure S8: Micrographs of spin-coated Au₃₂-NC thin films. (a) Optical micrograph of a 30 ± 2 nm thin film on interdigitated Au electrodes of a Si/SiO_x substrate. Optically, the thin film appears homogeneous over the entire displayed area of ~ 1.5 mm². Scale bar: 250 μ m. (b) Scanning electron micrograph of the same film within a channel of $L = 2.5$ μ m (Au electrodes at left and right side). A continuous film with individual grains of 440 ± 130 nm length can be observed. Scale bar: 500 nm. (c) High-resolution SEM micrograph of the Au₃₂-NC thin film. Scale bar: 100 nm. (d) SEM micrograph under incident angle of 85° of a thin film within a channel of $L = 2.5$ μ m (Au electrodes in dark, channel gap in bright). A smooth and uniform surface can be observed. Scale bar: 1.5 μ m.

5. Please add the axis name to the right side of Figure 4e.

The axis of Figure 4e has been updated in the revised manuscript (line 217).

6. Given the very low current intensity and high noise in Figure 4f, this data should be evaluated more carefully. The authors attributed the noise to “non-ideal channel geometry”, which is not that convincing to me, because the channel size is only needed for the post-measurement calculation of mobility, and should not affect the I-V curve that much during the measurement. Also, the authors showed the mean values of mobility and should also include the standard derivation.

We attribute the appearance of the FET transfer curve of an individual micro-crystal (Figure 4f) to the fact that the contact between the Au₃₂-NC crystal and the dielectric SiO₂ layer is not ideal. A schematic drawing is given below. The micro-crystals are deposited onto the electrodes with a thickness of ~10 nm. Accordingly, there is a gap between the micro-crystals and the dielectric of 0-10 nm. This is what we refer to as non-ideal channel geometry. In contrast, the spin-coated films form a relatively conformal layer within the channel, which leads to much better contact. Nonetheless, there is still an appreciable transconductance in the micro-crystals, which – after renormalization for the different channel geometry – is ~30 higher for the micro-crystals compared to the spincoated films. This further indicates the more efficient charge transport in highly ordered micro-crystals.

Figure S12: Schematic drawing of FET devices of a micro-crystal (left) and a thin film (right).

Details about the current normalization considering the different channel geometries:

- thin film: $I_{\text{film}}(-40 \text{ V}_G, 5 \text{ V}_{sd}) = 5.5 \times 10^{-8} \text{ A}$
- micro-crystal: $I_{\text{crystal}}(-40 \text{ V}_G, 5 \text{ V}_{sd}) = 9.0 \times 10^{-9} \text{ A}$

Normalized to geometry:

- thin film: $\frac{I_{\text{film}} \cdot L}{W \cdot h} = \frac{5.5 \times 10^{-8} \text{ A} \cdot 2.5 \times 10^{-6} \text{ m}}{0.01 \text{ m} \cdot 30 \times 10^{-9} \text{ m}} = 4.58 \times 10^{-4} \text{ A/m}$
- micro-crystal: $\frac{I_{\text{crystal}} \cdot L}{W \cdot h} = \frac{9.0 \times 10^{-9} \text{ A} \cdot 1.5 \times 10^{-6} \text{ m}}{10 \times 10^{-6} \text{ m} \cdot 100 \times 10^{-9} \text{ m}} = 1.35 \times 10^{-2} \text{ A/m}$

Ratio:

$$\frac{1.3 \times 10^{-2} \text{ A/m}}{4.5 \times 10^{-4} \text{ A/m}} = 29.5$$

This is included in the revised SI (Figure S12, line 843–861).

The mean value and standard deviation of the hole mobility from 21 individual micro-crystals can be calculated to be $\mu(h^+) = 0.8 \times 10^{-4} \pm 0.58 \times 10^{-4} \text{ cm}^2 \text{ V}^{-1} \text{ s}^{-1}$. Values up to $2 \times 10^{-4} \text{ cm}^2 \text{ V}^{-1} \text{ s}^{-1}$ are observed. The following figure has been included in the revised SI (Figure S11, line 834–841) and the standard deviation is given in the revised manuscript (line 277–279).

Figure S11: Distribution of the field-effect hole mobility $\mu(h^+)$ of 21 individual micro-crystals. All micro-crystals show p-type behavior. The mean value and standard deviation can be calculated to be $\mu(h^+) = 0.8 \times 10^{-4} \pm 0.58 \times 10^{-4} \text{ cm}^2 \text{ V}^{-1} \text{ s}^{-1}$. Values up to $2 \times 10^{-4} \text{ cm}^2 \text{ V}^{-1} \text{ s}^{-1}$ are observed.

7. In Abstract, the meaning of “structure-transport” is vague and confusing. In Line 22, it should “resulted from” not “resulting from”.

Thank you for this remark. The sentence “This first structure-transport correlation on self-assembled superstructures...” has been replaced by “This first correlation of structure and electronic properties by comparing glassy and crystalline self-assembled superstructures...” in the revised abstract (line 39–41). The suggested phrase is also changed to “resulted from” (line 35).

8. In Line 60 Page 3, “100-fold decrease” is not a correct expression. “1-fold decrease” meaning going from 100% to 0%. Here, it should be “decrease to 1%” if that’s what the authors were trying to say.

Thank you for this remark, which we have corrected in the revised manuscript accordingly (line 77–78, line 347–348).

9. Authors should make a broader intellectual framing for their work since the “transport” issue in the hierarchical materials is not limited to gold nanoclusters. Other cluster-based materials should be mentioned in the introduction. Some notable recent examples: Nature Mater. 2017, 16, 83-88, Nature Mater. 2017, 16, 474–480, Nature Mater. 2018, 17, 341-348.

We have implemented a broader intellectual framing on solid-state materials formed by molecular clusters in the introduction, including notable references (line 55, 58–61, references 11–20).

Reviewer #2 (Remarks to the Author):

In this article, Fetzer et al reported the electrical conductivity and charge transport behavior of the Au₃₂(Bu₃P)₁₂Cl₁₈ micro-crystals and films. The micro-crystals were formed by self-assembly at the liquid-air interface. A key conclusion is that the electrical conductivity of Au₃₂ micro-crystals is 100-fold higher than the Au₃₂ film (10⁻⁴ vs 10⁻⁶ S/m). Temperature-dependent conductivity measurements show that hopping activation energy is lower in micro-crystal than in the film (227 meV vs 366 meV), indicating the film is more disordered. The manuscript was well organized and the data was carefully presented. However, there are some major issues need to be further adjusted.

1. The authors suggested that the increased conductivity is due to the enhanced electronic coupling in micro-crystals, as highlighted in the article title. However, the lower conductivity of thin film could just be a result of cracks or inhomogeneity in the large area of spin-coated films, as suggested in Line 306-308. Lacking disorder in micro-crystal does not necessarily mean there is an electronic coupling between clusters. Also, there is no direct evidence or theoretical demonstration of enhanced electronic coupling such as the formation of extended band structures. Please consider adjusting the title to better reflect the key findings such as “enhanced electrical conductivity/charge transport...” instead of using the vaguely defined “electronic coupling”.

Thank you for suggesting an alternative title. We changed the title of the revised manuscript to “Structural order matters: Enhanced charge carrier transport in self-assembled Au-nanoclusters”.

2. Compared with characterizations on the micro-crystals, there is little information about the film quality, such as smoothness and uniformity. These are also critical data since the conductivity of the thin film is significantly affected by the quality of the film. The authors mentioned that the thickness of the film is only 30-50 nm, which is much thinner than the thickness of the micro-crystals (50-600 nm). The thinner film could have more defects such as pinholes. If the quality or the thickness of the film is increased, will the conductivity be increased accordingly?

Thank you for the note that our manuscript here requires additional data. In the revised manuscript and SI, we have included further details on the thin films and the following figures (Figure S8 and S9, line 764–794). From optical and SEM micrographs it is apparent that the spin-coated samples consist of homogeneous and continuous thin films. Individual grains of

440 ± 130 nm length can be observed. Using profilometry, the thickness was determined. Measuring the thickness on several positions of the samples yielded only small thickness deviations of 6.7–8.5% (30 ± 2 nm and 47 ± 4 nm). A root-mean-square roughness of 2.2 nm was measured, corroborating the comparably homogenous nature of the films.

Considering the already high quality of the spin-coated films, we do not expect a significant increase of the conductivity upon decreasing the roughness even further. At this point, the conductivity appears to be limited by the electronic properties of the individual clusters.

Figure S8: Micrographs of spin-coated Au₃₂-NC thin films. (a) Optical micrograph of a 30 ± 2 nm thin film on interdigitated Au electrodes of a Si/SiO_x substrate. Optically, the thin film appears homogeneous over the entire displayed area of ~ 1.5 mm². Scale bar: 250 μ m. (b) Scanning electron micrograph of the same film within a channel of $L = 2.5$ μ m (Au electrodes at left and right side). A continuous film with individual grains of 440 ± 130 nm length can be observed. Scale bar: 500 nm. (c) High-resolution SEM micrograph of the Au₃₂-NC thin film. Scale bar: 100 nm. (d) SEM micrograph under incident angle of 85° of a thin film within a channel of $L = 2.5$ μ m (Au electrodes in dark, channel gap in bright). A smooth and uniform surface can be observed. Scale bar: 1.5 μ m.

Figure S9: Thickness characterization of Au₃₂-NC thin films by profilometry. (a) Camera image of the profilometer showing an electrode device with a spin-coated Au₃₂-NC thin film and the stylus (as well as its reflection). The red arrow indicates the scanning direction x across a scratch within the thin film. Scale bar: 500 μm . (b) Corresponding height profile of the thin film revealing a thickness of $h = 30$ nm and a root-mean-square roughness of 2.2 nm.

3. For the absorption spectrum of micro-crystals, there is an additional very broad peak at 1.55 eV, which the author assigned as the HOMO-LUMO transition (line 162, 176-178). However, the additional peak can be a result of interference phenomena since the thickness of the crystal is in the same regime of Vis-NIR wavelength (line 107). Can the author provide any evidence to exclude this possibility?

To test this alternative explanation suggested by the reviewer, we investigated the absorbance spectra of eleven individual micro-crystals with different thicknesses, as displayed in the following Figure (a: absorbance spectra, b: energy-corrected absorbance spectra, c: optical micrographs). Here, the raw data as well as smoothed lines are plotted to emphasize the peak positions (all curves normalized to the local maximum at 480 nm / 2.58 eV). All eleven micro-crystals exhibit the enhanced peak at around 1.55 eV (800 nm). The intensity varies, caused by the thickness of the micro-crystals. Since this speaks against interference phenomena, where the additional peak is expected to appear at different wavelengths, due to the different thicknesses of the eleven micro-crystals. We believe that these additional measurements corroborate our interpretation as a material-specific property.

This figure has been included in the revised SI (Figure S4, line 692–703) and is referenced in the revised manuscript (line 198–199).

Figure S4: Absorbance spectra of individual Au₃₂-NC micro-crystals. **(a)** Absorbance as a function of wavelength λ of 11 micro-crystals on glass. **(b)** The spectra from (a) with energy corrected absorbance (using the expression $I(E) = I(\lambda) \times \lambda^2$) as a function of energy. All micro-crystals exhibit the enhanced absorbance peak at around 1.55 eV (800 nm). The raw data as well as smoothed lines are plotted to emphasize the peak positions (all curves normalized to the local maximum at 480 nm / 2.58 eV). **(c)** Optical micrographs (camera images) of the eleven micro-crystals where absorbance was measured (labeled). Scale bar corresponds to 40 μm and apply to all subfigures.

4. Line 290-292, it is not proper and meaningful to claim the first time. For example, in ref 18, the conductivity was measured on a single crystal. Consider remove or rephrase the sentence.

Thank you for pointing this out. We have rephrased this sentence it in the revised manuscript to: “While the seminal work by Li et al. reported electric transport measurements on similar single crystals for the first time, we provide a direct comparison of the transport properties in the ordered vs. the glassy state. This uniquely allows us to quantify the value of long-range order for electric transport in Au-NC ensembles.” (line 312–316)

5. Typo: line 252, mobility should be 10⁻⁶ to 10⁻⁵.

Thank you for pointing out that typo. It is corrected in the revised manuscript (line 274).

Reviewer #3 (Remarks to the Author):

In this manuscript, the authors reported the controlled self-assembly of atomically precise Au₃₂(nBu₃P)₁₂Cl₈ nanoclusters into micro-crystals. By comparing micro-crystal with dispersion and thin film, they demonstrated that the long-range order in micro-crystal enhances electronic coupling, resulting in significant enhancement of electric conductivity and charge carrier mobility. Generally, the data and analysis are comprehensive and solid. However, as despite the significant work done, I think its current version does not belong to Nature Communication, unless the authors can provide more solid measurement data to support their conclusion. Below are a few comments:

1. The crystal structures of micro-crystals and structures of thin films should be characterized clearly by X-ray single crystal diffractometer to determine how Au₃₂-NCs interact with neighboring clusters, which is critical for explaining why the electronic coupling is enhanced in micro-crystals relative to in thin films.

We have attached the X-ray analysis of the Au₃₂-cluster which was previously published by our group [*Angew. Chem. Int. Ed.* **58**, 5902–5905 (2019)]. Since the spin-coated films of Au NCs do not exhibit long-range order, they do not show any specific X-ray reflections, precluding a structural analysis. We did conduct GISAXS measurements for the amorphous films as well as for the crystals. From this, we concluded that the mean spacing between the single cluster cores is comparable for crystals as well as for the thin films. Therefore, the significant difference between both is the lack of long-range order in the thin films.

	Au₃₂(ⁿBu₃P)₁₂Cl₈ 1Bu
Formula	C ₁₄₄ H ₃₂₄ Au ₃₂ Cl ₈ P ₁₂
Formula weight	9014.18
T /K	150.0
Crystal system	triclinic
Space group	P $\bar{1}$
a /Å	19.083(4)
b /Å	19.337(4)
c /Å	33.249(8)
α /°	73.233(2)
β /°	86.735(3)
γ /°	63.435(2)
V /Å ³	10470(4)
Z	2
μ /mm ⁻¹	22.544
ρ /g·cm ⁻³	2.859
Reflns. meas.	241275
Independent reflns.	34860
R (int.)	0.1073
GooF	1.162
R _I (I > 2σ)	0.0699
wR ₂ (all data)	0.1160
CCDC number	1883285

2. How the Au₃₂-NCs assemble into micro-crystals? And what is the structure difference between single crystals and thin films. This could help to explain the claim that the electronic coupling of micro-crystals is enhanced comparing with thin films.

Single micro-crystals assemble at the surface of the liquid subphase, enabling crystallization. Afterwards they sink down onto the substrate, which is placed within the subphase. This is described in the SI (line 895–908). Thin films are created by spin-coating, inhibiting the formation of crystals, or - more accurately - long range order of the nanoclusters is inhibited while the nanocluster spacing remains almost the same (according to GISAXS). Accordingly, the micro-crystals exhibit long-range ordered NCs, whereas the NCs within the thin films are in a glassy–polycrystalline state.

3. The claims that "this first structure-transport correlation on self-assembled superstructures of atomically precise gold nanoclusters" and "however, attempts to quantify the influence of perfect order on the electronic properties of such micro-crystals have remained unsuccessful" are not objective enough since reference 18 has already reported the structure-transport correlation of Au NPs single crystal.

Thank you for pointing this out. We have rephrased this sentence it in the revised manuscript to: "While the seminal work by Li et al. reported electric transport measurements on similar single crystals for the first time, we provide a direct comparison of the transport properties in the ordered vs. the glassy state. This uniquely allows us to quantify the value of long-range order for electric transport in Au-NC ensembles." (line 312–316)

4. Please try to correlate the semiconductor characteristic and hole-conduction mechanism with the unique structure of the micro-crystals.

We assume thermal activation to be responsible for the conductivity in the Au₃₂-NC ensembles. This is concluded based on the high temperature dependency of the resistance of the films. The semiconducting behavior of the system is ensured by the organic tri-*n*-butyl phosphine ligands, limiting the electronic coupling between the cluster cores just enough to allow conductance without entering metallic band formation. The importance of controlling the cluster core distance to ensure semiconducting behaviour was already shown by previous studies [*Adv. Mater.* **31**, 1900684 (2019)].

These studies further assumed that the n-type conductivity of their gold nanocluster films arises from trapping of hole charge carriers due to the presence of Cl⁻ and Br⁻ anions in their films and therefore favouring the electron conductivity. We present uncharged clusters as building blocks which show p-type behaviour, verified by FET measurements. This is in accordance with the reports of Yuan et al. who also investigated p-type conductivity for their films built of uncharged Au/Ag₃₄ nanoclusters [*Nat. Commun.* **11**, 2229 (2020)].

To gain further insights, a detailed consideration of the overlapping of the electronic states between neighbouring cluster cores would be required which is, however, far beyond the scope of this paper.

5. Since the micro-crystals has strongly preferred growth direction, it is better to compare the electrical properties of different crystallographic axes.

Thank you for this highly interesting suggestion. We have investigated the crystal axis dependent conductivity of individual micro-crystals, which is detailed below.

The favored growth direction of the Au₃₂-NC micro-crystals is along the *a* and *b* axes. They are almost identical (1.90 nm and 1.94 nm, respectively) and cannot be distinguished in the micro-crystals. The crystal-growth along the larger *c* axis (3.48 nm) is unfavored causing the micro-crystals to have thicknesses of only ~50–600 nm, whereas the lateral expansions are in the range of ~5–30 μm. The micro-crystals are lying face down with the *c* axis out of plane. Thus, no conductivity measurements of micro-crystals along the crystal axis *c* could be performed.

We investigated the electronic conductivity of individual microcrystals as a function of their azimuthal orientation, displayed in the following figure. We have defined the azimuthal angle θ as the relative angle between the long axis of the micro-crystal (fig. a, white line) and the electrode edge. For symmetry reasons, the azimuthal angle θ can rotate from 0 to 90°.

Although there is no obvious trend in conductivity as a function of θ (figure b), the data points of the individual micro-crystals are not completely randomly scattered. A local maximum between $\theta = 40\text{--}60^\circ$ seems to be present, which is the around the expected angle of $\theta = 56^\circ$ where the crystallographic axis (*a* or *b*) is parallel to the electric field vector (figure c, configuration 2). This might be a first hint for an anisotropic charge transport in Au₃₂-NC microcrystals. For azimuthal angles reaching 90°, the micro-crystals channels can have two different configurations (configuration 3 and 5, figure c), which further complicates the investigation.

However, a careful further exploration would be beyond the scope of this paper, which is focused on the differences in charge transport between ordered and glassy Au-NC ensembles. We would like to thank the reviewer for the hint to look into the direction dependent conductivity. Nonetheless, we prefer not to include these measurements in the present manuscript, as this might distract from the actual focus of the evaluation of the importance of order on transport behavior.

6. The conductivity of micro-crystal is tested with two-point probe method. It will be better to conduct a 4-point probe measurement for this system. This is recommended for measuring electrical conductivity in metal-organic framework materials. See: *J. Am. Chem. Soc.* 2016, 138, 44, 14772-14782.

We agree that a 4-point probe measurement would be ideal to assess and measure the contact resistance. However, as stated in the reference mentioned by the reviewer, “due to the small size of single crystals of MOFs (<1 mm), methods involving four contacts can be challenging.” Our crystals fall into this size regime, and we have indeed not been able to create four reliable contacts. Therefore, we tried a different approach that we feel is more suitable for the small samples analysed in this work, which will still allow for a determination of the contact resistance.

Our first attempt in this direction is an application of the transmission line method (TLM) by measuring the total resistance for varying channel lengths ($L = 2.5 - 20 \mu\text{m}$). Linear fitting the measured resistance as a function of channel length yields R_C (y-axis intercept) [Zaumseil et al., *J. Appl. Phys.* **93**, 6117–6124 (2003)]. We find a y-intercept close to zero, indicating that R_C is negligible (see Figure below). However, the slightly negative value indicates that the TLM may not be fully reliable here. This is probably due to an increasingly non-ohmic I - V characteristic for largest channel length (20 μm), which is also reflected by the large error bar.

Figure: Transmission line method (TLM) to estimate the contact resistance R_C . Mean values and standard deviations of several devices for each channel length are displayed.

Therefore, we also apply the Y-function method to measure R_C [Ghibaudo, *Electron. Lett.* **24**, 543 (1988); Xu et al., *J. Appl. Phys.* **107**, 114507 (2010); Liu et al., *Mater. Today* **18**, 79–96 (2015)]. This is included in the revised SI (Figure S13, line 867–892).

The following figure illustrates the YFM technique to estimate the R_C of a FET device (exemplarily shown for an Au_{32} -NC thin film channel).

Figure S13: Y-function method (YFM) to estimate the contact resistance R_C of an individual FET device. (a) Transfer characteristic of a Au₃₂-NC thin film p-type FET at $V_{sd} = 1$ V ($L = 2.5$ μm , $W = 1\text{cm}$). (b) Transconductance versus gate voltage V_G . (c) Y-function versus V_G . Fitting the linear regime yields the slope $s1$. (d) $1/\sqrt{g}$ versus V_G . Fitting the linear regime yields the slope $s2$. The R^2 values in (c) and (d) verify the goodness of the linear fit. R_C is calculated to 1.2×10^7 Ω . The total resistance of this device is 1.9×10^9 Ω .

From the transfer characteristic (I_{SD} vs. V_G) the transconductance g is determined as $g = \partial I_{SD} / \partial V_G$.

The Y-function is defined as

$$Y = \frac{I_{SD}}{\sqrt{g}}$$

and fitting Y as a function of V_G in the linear regime yields the slope $s1$ (figure c). Next, the function $\frac{1}{\sqrt{g}}$ versus V_G is determined and linearly fitted to calculate the slope $s2$ (figure d). The contact resistance R_C is calculated as

$$R_C = V_{SD} \times \frac{s2}{s1} .$$

Using this method, we determine the contact resistance of thin films and microcrystal channels as $R_{C,\text{films}} \approx 1\text{--}3 \times 10^7$ Ω and $R_{C,\text{crystals}} \approx 2 \times 10^8$ Ω , respectively. In contrast, the total resistances are $R_{\text{films}} \approx 0.5\text{--}2 \times 10^9$ Ω and $R_{\text{crystals}} \approx 2\text{--}20 \times 10^9$ Ω . Thus, the contact resistances are only $\sim 2\%$ and $\sim 1\text{--}10\%$ for thin film and microcrystal devices, respectively.

Thus, the Y-function method verifies the applicability of simple 2-point-probe measurements, as the effect of R_C is negligible. Generally, the 2-point probe method is applicable for conductive materials with resistances higher than 1 k Ω , as the typical resistance of contacts and wires is around 100 Ω (J. Am. Chem. Soc. 2016, 138, 44, 14772-14782). The Au₃₂-NC micro-crystals presented in the manuscript have a typical resistance of 2–20 G Ω , which is 6–7 orders of magnitude larger than the critical lower limit value for 2-point probe methods.

7. Figure 1b-d and Figure 4a should be marked clearly.

We have updated the figure captions accordingly in the revised manuscript (line 89–91, line 219–222).

REVIEWERS' COMMENTS

Reviewer #1 (Remarks to the Author):

Authors have conducted an incredibly thorough revision of their original manuscript addressing the majority of questions posed. I enthusiastically recommend publication in the current form.

Reviewer #2 (Remarks to the Author):

The authors have addressed most of my concerns in the revised manuscript. The answers and additional data lead to further questions. For example, what is the packing density of the clusters in the film compared with that in the crystal? Although it is smooth, the film does not seem to pack densely (Figure S8 c). Small void (5-10 nm) can be seen in the SEM image of the film. The decrease of conductivity in the film compared with crystal can be a result of the lower packing density.

Also, in Figure S4, why is the intensity of 800 nm peak changes with the thickness of the crystal while the intensity of 480 nm peak remains the same? Is it possible to measure the thickness and the absorbance spectra of the microcrystal at the same time? This correlation may provide more insight into the emergence of the 800 nm peak in crystal.

Despite these minor questions, a lot of effort has been made and the manuscript has improved compared with the first version. Therefore, the current version is suitable for publication in Nature Communication.

Reviewer #3 (Remarks to the Author):

The authors have generally addressed the comments of the reviewers. I recommend that the manuscript is accepted.

Reviewer comments are denoted in black.

Author replies are denoted in green.

Specific changes in the manuscript are denoted in red. Indicated line numbers correspond to the revised manuscript.

REVIEWER COMMENTS

Reviewer #1 (Remarks to the Author):

Authors have conducted an incredibly thorough revision of their original manuscript addressing the majority of questions posed. I enthusiastically recommend publication in the current form.

Thank you very much for the positive feedback. We really appreciate the success of this review process and think that the quality of the paper was incredibly improved by your comments and suggestions.

Reviewer #2 (Remarks to the Author):

The authors have addressed most of my concerns in the revised manuscript. The answers and additional data lead to further questions. For example, what is the packing density of the clusters in the film compared with that in the crystal? Although it is smooth, the film does not seem to pack densely (Figure S8 c). Small void (5-10 nm) can be seen in the SEM image of the film. The decrease of conductivity in the film compared with crystal can be a result of the lower packing density.

Thank you for this remark. Indeed, the nanocluster packing density within micro-crystals and thin films could differ. However, a difference in conductivity by a factor of 100 cannot be explained solely by the packing density, as long as the volume fraction of the thin film is still above the percolation threshold. This threshold is approximated with 20% (Phys. Rev. B 1980, 21, 3725), and we can clearly deduce from our electron microscopy images that the volume fraction is substantially above this value for the thin films and the micro-crystals. In such a situation, the resistance of percolative networks is usually not dramatically reduced by increasing the packing density.

Also, in Figure S4, why is the intensity of 800 nm peak changes with the thickness of the crystal while the intensity of 480 nm peak remains the same? Is it possible to measure the thickness and the absorbance spectra of the microcrystal at the same time? This correlation may provide more insight into the emergence of the 800 nm peak in crystal.

Thank you for this suggestion, which we followed up. The following figure has been included in the revised SI (Figure S4, line 748–762). We qualitatively determined the thickness of the corresponding micro-crystals on which the absorbance spectra were measured using the grayscale value from the camera image. This thickness parameter is now included in the optical spectra via the color scale legend. We observe that the emerging 800 nm peak is more pronounced for thinner micro-crystals. This could be explained by the fact that thinner micro-crystals are more defect-free, while for thicker crystals cracks and inhomogeneities are often observed (e.g. thin micro-crystal in Figure 4a vs. thick micro-crystal in Figure S6), which in turn might adversely affect the electronic coupling. An alternative, additional explanation are the challenges involved with finding the focal plane for thick micro-crystals, which may result in an artificially reduced absorption intensity.

Figure S4: Absorbance spectra of individual Au₃₂-NC micro-crystals. (a) Absorbance as a function of wavelength λ of eleven micro-crystals on glass. The legend indicates the qualitative thickness of the corresponding micro-crystals, determined via the grayscale values from the optical micrographs in (c). (b) The spectra from (a) with energy corrected absorbance (using the expression $I(E) = I(\lambda) \times \lambda^2$) as a function of energy. All micro-crystals exhibit the enhanced absorbance peak at around 1.55 eV (800 nm). All curves are normalized to the local maximum at 2.58 eV (480 nm). (c) Optical micrographs (camera images) of the eleven micro-crystals where absorbance was measured (labeled). From top left to bottom right, the grayscale value of the micro-crystals decreases, assuming the thickness to decrease as well. The scale bar corresponds to 40 μm and applies to all subfigures.

Despite these minor questions, a lot of effort has been made and the manuscript has improved compared with the first version. Therefore, the current version is suitable for publication in Nature Communication.

Thank you very much for your feedback. We value the success of this review process and think that the quality of the paper was immensely improved by your remarks and suggestions.

Reviewer #3 (Remarks to the Author):

The authors have generally addressed the comments of the reviewers. I recommend that the manuscript is accepted.

Thank you very much for your feedback. We are grateful for your feedback which has really improved the content of this publication.